# Dupilumab Improves Pruritus in Netherton Syndrome: A Case Study

**DOI:** 10.3390/children9030310

**Published:** 2022-02-24

**Authors:** Yutaka Inaba, Nobuo Kanazawa, Kyoko Muraoka, Azusa Yariyama, Ami Kawaguchi, Kayo Kunimoto, Chikako Kaminaka, Yuki Yamamoto, Kaoru Tsujioka, Akira Yoshida, Teruki Yanagi, Masatoshi Jinnin

**Affiliations:** 1Department of Dermatology, Wakayama Medical University, Wakayama 641-0012, Japan; kyoko_m@wakayama-med.ac.jp (K.M.); snoopy_and_woodstock_714_ay@yahoo.co.jp (A.Y.); amik@wakayama-med.ac.jp (A.K.); k-jigen@wakayama-med.ac.jp (K.K.); kamikami@wakayama-med.ac.jp (C.K.); yukiy@wakayama-med.ac.jp (Y.Y.); jinjin1011@hotmail.com (M.J.); 2Department of Dermatology, Hyogo College of Medicine, Nishinomiya 663-8501, Japan; 3Department of Dermatology, Japanese Red Cross Society Wakayama Medical Center, Wakayama 640-8558, Japan; qqmw91156@ares.eonet.ne.jp; 4Department of Pediatrics, Japanese Red Cross Society Wakayama Medical Center, Wakayama 640-8558, Japan; jxmcy423@yahoo.co.jp; 5Department of Dermatology, Hokkaido University, Sapporo 060-8638, Japan; yanagi@med.hokudai.ac.jp

**Keywords:** Netherton syndrome, dupilumab, atopic dermatitis

## Abstract

The patient was a 26-year-old male. He had red and scaling skin of the entire body since birth, as well as an elevated level of serum IgE. Genetic testing revealed a mutation in the SPINK5 gene, which had confirmed the diagnosis with Netherton syndrome. He has had significant pruritis since birth, and subsequently had symptoms of sleeping disorders and concentration difficulty throughout the day. Since treatment with various antihistamines were not effective, we administered dupilumab and found that it was effective in immediate elimination of pruritus and gradual reduction of the rash. Dupilumab has been administered for one year without any adverse events or recurrence of symptoms. Although studies have previously described cases who used dupilumab for Netherton syndrome, reported effects have been limited or transient. Additional studies are needed to confirm the effect of dupilumab for Netherton syndrome, which currently lack any effective treatment strategies.

## 1. Introduction

Netherton syndrome (NS) is an autosomal recessive disease characterized by generalized redness, scaling, and peeling of the skin epidermis throughout the entire body at or early after birth. It is also characterized by abnormal hair and an elevated IgE level. Mild NS is often misdiagnosed as atopic dermatitis; thus, genetic testing is useful for a definitive diagnosis. In the present study, we describe a case of NS who was treated by dupilumab and subsequently demonstrated significant improvement in pruritus and rash.

## 2. Case Presentation

The patient was a 26-year-old male who was clinically diagnosed with non-bullous congenital ichthyosiform erythroderma at birth (Figure 1a,b). He had been treated by a pediatrician and was subsequently referred to us in 2019 when he reached the age of 24. When he was 6 months old and 24 years old, skin biopsies from his trunk were performed. Both samples histopathologically showed psoriasiform hyperplasia with parakeratosis (Figure 2a,b). In 2019, genetic testing was performed and a homozygous frameshift mutation c.377_378delAT was identified in SPINK5. This led to the definitive diagnosis of NS. None of his family members had NS. 

Despite receiving treatments with moisturizers and antihistamines since birth, there was no improvements in the symptom of rash. Due to significant rash and pruritis, we decided to administer dupilumab, which became available for use in patients with atopic dermatitis, and whose effect on NS has been demonstrated in several studies [1,2,3,4,5]. Based on the recommended dose for atopic dermatitis, dupilumab was subcutaneously administered for the patient by an initial dose of 600 mg, followed by 300 mg every two weeks. 

The administration of dupilumab resulted in reduced redness and erythema of the face and the trunk, with only faint erythema remaining after one year (Figure 3). The numerical rating scale (NRS) for pruritus decreased to 0 soon after the administration of dupilumab. Similarly, the Eczema Area and Severity Index (EASI) score for atopic dermatitis decreased gradually (Figure 4a). A blood test revealed that the IgE level and eosinophil counts both decreased soon after the administration of dupilumab. There was no significant change in thymus and activation-regulated chemokine (TARC), which was at a relatively low level prior to the administration of dupilumab (Figure 4b). 

The patient had been on the treatment for one year without any adverse events or recurrence of symptoms. 

## 3. Discussion

In the present study, we demonstrated the case of NS who was treated with dupilumab. The treatment induced an immediate elimination of pruritis and gradual reduction of rash, without any recurrence of symptoms or any adverse events associated with the treatment for one year. 

In NS, the frequency, duration, and severity of itch are comparable to that of patients with atopic dermatitis, and the patient’s lives are significantly affected by pruritus. In the other subtypes of ichthyosis (including keratinopathic ichthyoses, autosomal recessive congenital ichthyoses, X-linked recessive ichthyoses, Sjögren–Larsson syndrome, and loricrin ichthyosis), the parameters associated with itch appeared to be less severe [6]. In the present study, we demonstrated that dupilumab was immediately effective in improving pruritus and the overall quality of life of the patient who had previously been treated unsuccessfully with various antihistamines. NS may be misdiagnosed as atopic dermatitis given the similar clinical presentations. A study also demonstrated that NS overlaps with atopic dermatitis due to dysregulated protease activities [7].

Notably, a study reported that peripheral blood T cells from NS patients stimulated with 12-O-tetradecanoyl phorbol-13-acetate and Ca2^+^ ionophore released more IL-4 and IL-13 compared with the healthy control [8]. In addition, Oetjen reported that type IL-4 and IL-13 are the master regulators of chronic itch [9]. Since dupilumab was effective in improving the symptoms of pruritus in the NS patient, our findings suggest that IL-4 and IL-13 are also involved in inducing pruritus in NS, as in atopic dermatitis. 

In addition, upregulation of IL-33 was detected in the epidermis of NS patients [10]. These results suggest that the pathophysiology of NS is shared with that of atopic dermatitis. 

To our knowledge, a total of eight case studies including our present study demonstrated the positive effect of dupilumab in NS patients (Table 1). Two of the studies reported recurrence of pruritus two weeks after the administration of dupilumab, and one of them further demonstrated a worsening of rash after eight weeks. The administration of dupilumab was terminated in one case due to resistance. For the three cases who presented the resistance, it was not mentioned whether they had neutralizing antibodies or not. These cases were all females; indeed, it is inconsistent with previous reports that women with atopic dermatitis had a good treatment response by dupilumab [11,12]. NS patients perhaps have different reasons why female patients tend to have resistance to dupilumab, but the number of NS patients who were treated with dupilumab is limited. A large number of NS patients is needed to confirm the reason why these patients showed the resistance. 

In the present study, we demonstrated that dupilumab was effective in immediately eliminating the symptom of pruritus in a patient who had been suffering from pruritus since birth. As a result, the patient experienced the lack of pruritus for the first time in his life and achieved high satisfaction by the treatment.

On the other hand, histopathological features of the patient’s skin lesion resembled that of psoriasis vulgaris (Figure 2). Claire and colleagues reported that the skin lesions in NS cases share the IL-17/IL-36 signature [13]. In addition, several papers reported that secukinumab (interleukin-17A inhibitor) was effective for NS treatment [14,15]. Collectively, secukinumab might be also effective in our case.

NS currently lacks any effective treatment strategies. We demonstrated that dupilumab was effective in treating the symptoms of NS. However, given that it did not result in a treatment response in some previous cases, all the patients who showed resistance for dupilumab are female. However, our case is the eighth NS patient treated with dupilumab, and a large number of patients treated with dupilumab is needed to detect why some NS patients are resistant to dupilumab. NS patients showing resistance might have neutralizing antibodies. At least in some NS patient, including our case, dupilumab was effective. NS is an orphan disease with multisystemic and severe complications, with currently no specific treatment available. Therefore, an unmet need exists for effective treatment. Dupilumab may lead to the development of therapies for controlling NS patients.

## Figures and Tables

**Figure 1 children-09-00310-f001:**
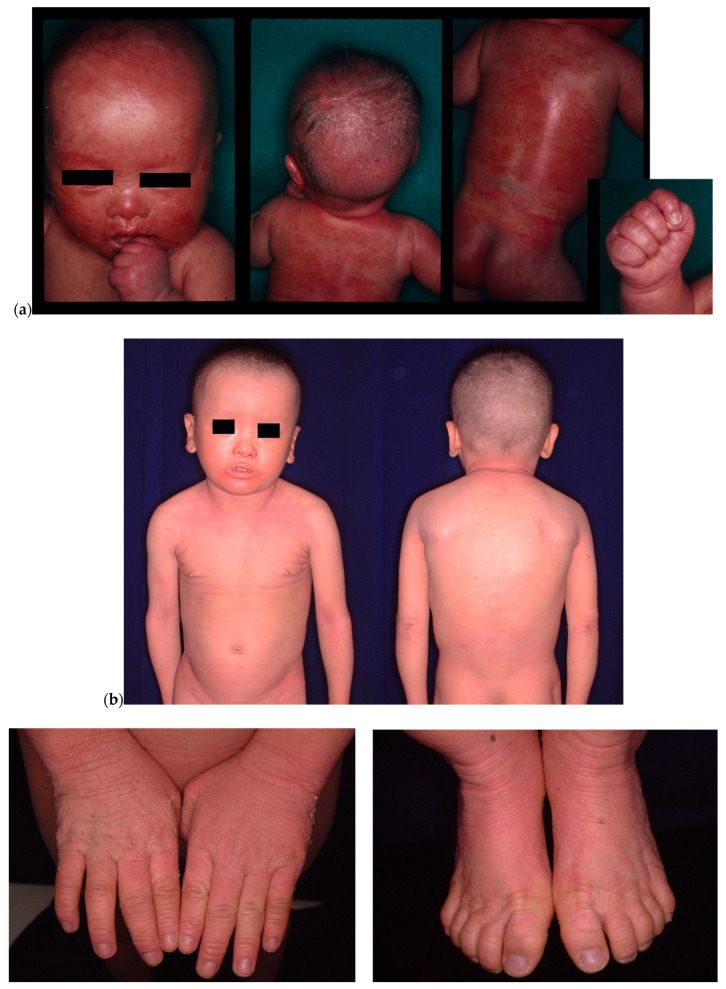
(**a**) Clinical images when the patient was six months old. Erythema on the whole body and scales on his head were detected. (**b**) Clinical images when the patient was five years old. Slight erythema on the whole body was still detected.

**Figure 2 children-09-00310-f002:**
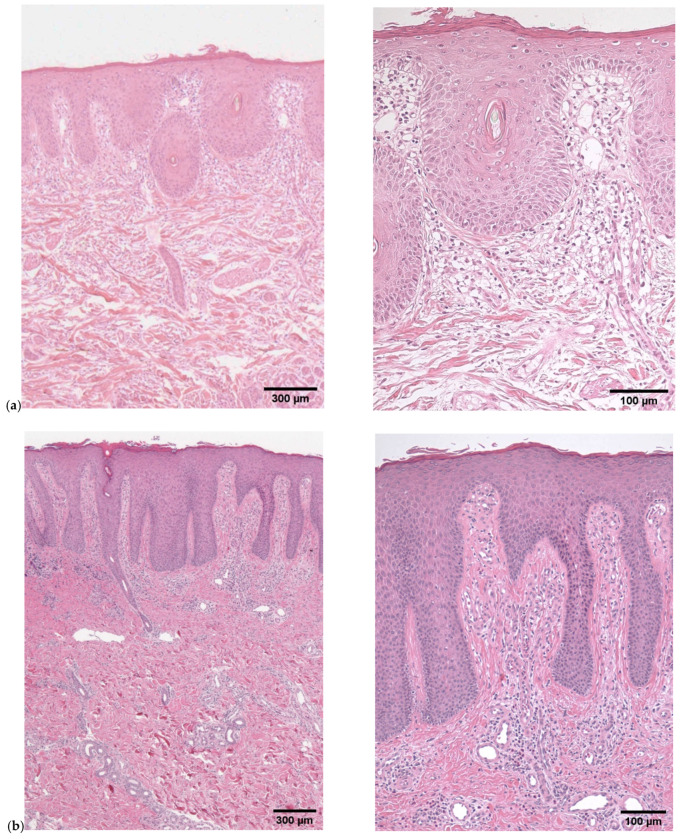
(**a**) Image of histopathology from trunk when he was six months old. Infiltrated lymphocytes, edema in the upper dermis, parakeratosis, and psoriasiform hyperplasia were detected. (**b**) Image of histopathology from trunk when he was 24 years old. Infiltrated lymphocytes in the upper dermis, parakeratosis, and psoriasiform hyperplasia were detected.

**Figure 3 children-09-00310-f003:**
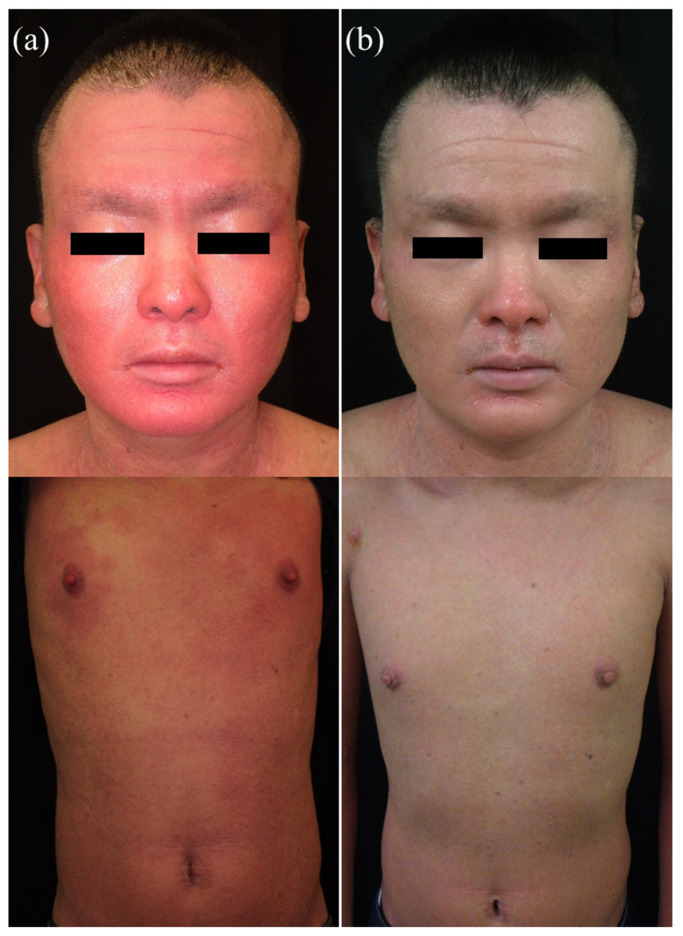
Clinical images (**a**) before and (**b**) one year after the treatment. The treatment was effective in reducing redness and erythema on the face and trunk.

**Figure 4 children-09-00310-f004:**
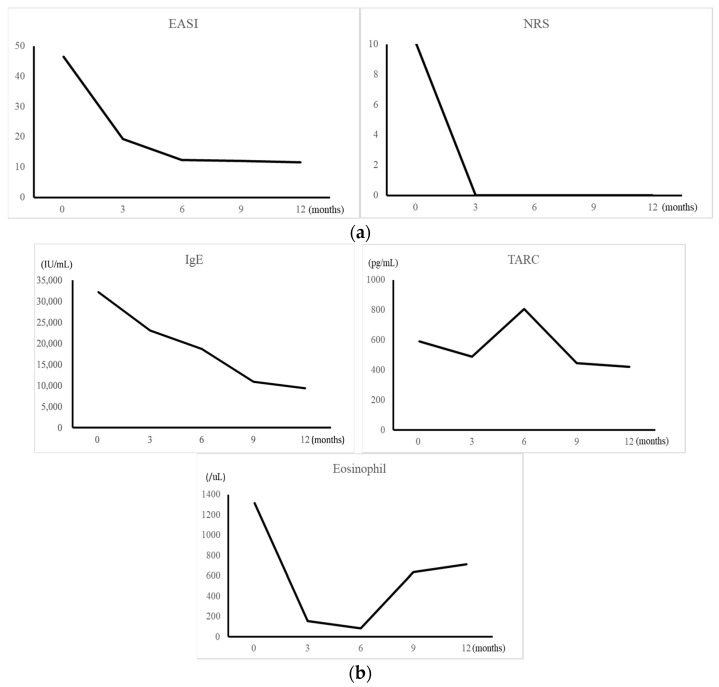
(**a**). Changes in the numerical rating scale (NRS) for pruritus and the Eczema Area and Severity Index (EASI) for atopic dermatitis before and after the administration of dupilumab. The x-axis indicates the elapsed time from the administration of dupilumab; the y-axis indicates EASI or NRS. (**b**). Changes in serum IgE, thymus and activation-regulated chemokine (TARC) and eosinophil before and after the administration of dupilumab. The x-axis indicates the elapsed time from the administration of dupilumab; the y-axis indicates IgE, TARC or eosinophil.

**Table 1 children-09-00310-t001:** List of previously reported cases of Netherton syndrome treated with dupilumab. A total of eight cases (including the present case) have been reported to date. Three patients developed resistance to dupilumab.

Author	Sex	Age	Treatment Duration (Months)	Response	Adverse Effects
Steuer [1]	female	32	18 M	Sustained	None
Andreasen [2]	male	43	6 M	Sustained	Not reported
Süßmuth [3]	female	12	12 M	Sustained	Bacterial infection
Süßmuth [3]	male	8	10 M	Sustained	Not reported
Aktas [4]	female	40	3 M	Temporary	Conjunctivitis
Murase [5]	female	32	6 M	Temporary	None
Murase [5]	female	17	6 M	Temporary	None
Present case	male	26	12 M	Sustained	None

M: month.

## Data Availability

The data presented in this study are available on request from the corresponding author. The data are not publicly available due to privacy.

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
