# Peer review of "Dupilumab Improves Pruritus in Netherton Syndrome: A Case Study"

_children, 2022, doi:10.3390/children9030310_

Round 1
Reviewer 1 Report
Thank you for asking me to review this extremely interesting and well written case study. It was a pleasure to read and my comments are minor:
- As this case study involves and adult patient, it would be worth including a paragraph or two highlighting the relevance of this case study to the paediatric population, given this journal focuses on that age group.
- Line 102 - reads 'a total of 8 studies' - I think these are all case studies and this should be made clear.
- The heading of the table includes text to state that 'resistance' developed in some cases. Was this resistance definitive, i.e. were neutralizing antibodies detected? It would be good to make this clear, to highlight whether this was true resistance or rather a lack of effect.
- Paragraph at line 115 - The significance of this paragraph isn't particularly clear. It would be good to develop the argument a little further.
- The final paragraph at line 119 feels a little underdeveloped which is a shame given that there is currently no effective treatment for NS and Dupilumab has shown promise. As this is the final paragraph it would be good to develop the argument further highlighting that case studies show promise for some but not others. Rather than stating simply that further studies are needed, perhaps you could discuss what these studies might look like/the mechanistic aspects that could be explored in them to establish definitively whether dupilumab is an effective treatment for this orphan disease.
Author Response
Thank you for giving me comments. I modified my paper as you pointed.
Please check my answer below.
1, As this case study involves and adult patient, it would be worth including a paragraph or two highlighting the relevance of this case study to the paediatric population, given this journal focuses on that age group.
Our case was treated with dupilumab when he was 26 years old. But he suffered from itchiness due to netherton syndrome since his birth. In some juvenile patients, dupilumab was effective. That is why we submit our paper for your article. It is our pleasure to let pediatric physician to know the effectiveness of dupilumab for netherton syndrome.
2,Line 102 - reads 'a total of 8 studies' - I think these are all case studies and this should be made clear.
I modified ‘a total of 8 case studies’ instead of ‘a total 8 studies’ in line104.
3,The heading of the table includes text to state that 'resistance' developed in some cases. Was this resistance definitive, i.e. were neutralizing antibodies detected? It would be good to make this clear, to highlight whether this was true resistance or rather a lack of effect.
Three cases who showed resistance for dupilumab were not mentioned whether they had neutralizing antibodies or not. That is why it is difficult to conclude the reason why these patients showed the resistance. But only female patients showed the resistance. I added the consideration in line 108 to 114.
4,Paragraph at line 115 - The significance of this paragraph isn't particularly clear. It would be good to develop the argument a little further.
Histopathology of skin biopsy in our case resemble that of vulgaris psoriasis. So we have a hypothesis that netherton syndrome have the same pathogenesis of vulgaris psoriasis. Some paper reported that netherton syndrome share IL-17/IL-36 signature and secukinumab (interleukin-17A inhibitor) was effective for netherton syndrome. That is why we expect that secukinumab is effective for our case.
5,The final paragraph at line 119 feels a little underdeveloped which is a shame given that there is currently no effective treatment for NS and Dupilumab has shown promise. As this is the final paragraph it would be good to develop the argument further highlighting that case studies show promise for some but not others. Rather than stating simply that further studies are needed, perhaps you could discuss what these studies might look like/the mechanistic aspects that could be explored in them to establish definitively whether dupilumab is an effective treatment for this orphan disease.
I added the discussion from line130 to 137. Even though the number of netherton syndrome treated with dupilumab is limited, only female netherton syndrome patient showed the resistance for dupilumab.
Best regards
Reviewer 2 Report
Inaba et al. presented an interesting case with Netherton syndrome (NS) that exhibited significant improvement following dupilumab injections. The clinical data is comprehensive and the language is clear. I have a couple of minor suggestions as follows.
- In the literature review, three patients showed temporary response to dupilumab. Why? Is there any difference between responsive patients vs. non-responsive ones? Could authors expend the discussion on when will be the best time to start dupilumab in NS?
- In my viewpoint, the most significant paper that could prove the effect of dupilumab in controlling itch was published by the Kim lab in Cell (2017, DOI: 10.1016/j.cell.2017.08.006). This should be mentioned or at least listed as a reference.
- Could authors provide informed consent from the patient to show that he agrees to publish his pictures?
Author Response
Thank you for giving me comments. I modified my paper as you pointed.
Please check my response.
1,In the literature review, three patients showed temporary response to dupilumab. Why? Is there any difference between responsive patients vs. non-responsive ones? Could authors expend the discussion on when will be the best time to start dupilumab in NS?
Three cases who showed resistance for dupilumab were not mentioned whether they had neutralizing antibodies or not. That is why it is difficult to conclude the reason why these patients showed the resistance. But only female patients showed the resistance. I added the consideration in line 108 to 114.
2,In my viewpoint, the most significant paper that could prove the effect of dupilumab in controlling itch was published by the Kim lab in Cell (2017, DOI: 10.1016/j.cell.2017.08.006). This should be mentioned or at least listed as a reference.
I cited the paper in line 98.
3,Could authors provide informed consent from the patient to show that he agrees to publish his pictures?
I will submit informed consent from the patient to the Journal office.